# Correction of Significant Urethral Anomalies Using a Tissue-Engineered Human Urethral Substitute: Proof of Concept

**DOI:** 10.3390/ijms26051825

**Published:** 2025-02-20

**Authors:** Christophe Caneparo, Elissa Elia, Stéphane Chabaud, François Berthod, Julie Fradette, Stéphane Bolduc

**Affiliations:** 1Centre de Recherche en Organogenèse Expérimentale de l’Université Laval/LOEX, Centre de Recherche du CHU de Québec-Université Laval, Axe Médecine Régénératrice, Québec, QC G1J 1Z4, Canada; christophe73.caneparo@gmail.com (C.C.); elissa.elia@crchudequebec.ulaval.ca (E.E.); stephane.chabaud@crchudequebec.ulaval.ca (S.C.); francois.berthod@fmed.ulaval.ca (F.B.); julie.fradette@fmed.ulaval.ca (J.F.); 2Department of Surgery, Faculty of Medicine, Université Laval, Québec, QC G1V 0A6, Canada

**Keywords:** urethra, urethroplasty, tissue engineering, oral mucosa, graft, rabbit

## Abstract

Urethral reconstruction remains a challenge. Indeed, the use of oral mucosa, the reference biomaterial for urethroplasty, is associated with two main drawbacks: the limited availability of autologous tissues and potential short- and long-term complications, especially for patients with recurrences or severe anomalies. Therefore, the development of alternative approaches, such as urethral tissue engineering, is necessary. A new type of human urethral substitute devoid of exogenous biomaterials has been reconstructed in vitro. It presented sufficient mechanical strength and had histological and functional properties comparable to native tissues. These reconstructed tissues were implanted in vivo to repair hypospadias induced in tacrolimus-immunosuppressed rabbits via a two-stage urethroplasty. In the first stage, the distal part of the native urethra was removed, and a flat graft was implanted, leaving the urethra open proximally. Twelve weeks later, the graft was tubularized to create a neourethra, reproducing the usual clinical scenario. The results obtained for the experimental group were less effective than for the control group, with a success rate of 50% after excluding the animal affected by unwanted events unrelated to urethroplasty, and it is possible that the animal model or surgical technique used was not suitable and should be modified. Nevertheless, half of the urethral substitutes grafted on rabbits showed successful integration. These self-assembled artificial tissues represent promising substitutes for urethroplasty.

## 1. Introduction

Penile reconstruction remains a significant challenge today, largely due to the limited availability of suitable tissues for urethroplasty and the frequent suboptimal outcomes, especially in cases involving recurrences or severe anomalies [1,2]. Among the most common penile pathologies is hypospadias, which affects approximately 1 in 250 boys. Its prevalence is on the rise, likely due to environmental endocrine disruptors [3,4,5]. Hypospadias is characterized by the abnormal positioning of the urinary meatus along the ventral side of the penis, with the most severe form, proximal hypospadias, where the meatus is located near the scrotum. In such cases, the urethral defect may extend over several centimeters or even involve the entire penile urethra. Unfortunately, there are limited tissue options for urethral repair or replacement. Various materials have been tested, including genital and extragenital skin grafts, *tunica vaginalis*, and oral or lingual mucosa, with oral mucosa remaining the typically used material when foreskin is unavailable [6,7,8,9,10,11,12,13,14,15]. However, its use remains challenging, as it can lead to a range of short- and long-term complications [13,16,17]. Additionally, the limited amount of oral mucosa that can be harvested further restricts therapeutic options, particularly for patients with recurrent conditions or severe anomalies. These limitations have driven researchers to explore alternative solutions.

Tissue engineering has introduced the use of several exogenous biomaterials aimed at restoring the structure and function of damaged tissues. Among these, synthetic, natural, and hybrid biomaterials have been investigated for urethral reconstruction [18,19,20,21,22,23]. Unfortunately, these materials often result in long-term complications and adverse clinical outcomes [8,24,25,26,27,28,29]. The urothelium that develops on many of those exogenous biomaterials exhibits limited maturation, with minimal or absent expression of key markers of terminal differentiation of the urothelial cells, which can be found in the watertight epithelium of the proximal urethra, such as mature uroplakin plaque, leading to reduced functionality. This is particularly concerning because the urothelium’s barrier function is critical in protecting underlying tissues from the cytotoxic effects of urine.

Using the self-assembly approach of tissue engineering, a functional human urethral substitute composed of vesical and dermal fibroblasts was successfully created [30]. This substitute is devoid of exogenous materials, as fibroblasts were stimulated to secrete and organize their own endogenous extracellular matrix (ECM) upon ascorbic acid stimulation, mimicking the natural stromal environment. The addition of urothelial cells on the substitute surface followed by 21 days of maturation at the air-liquid interface further enhanced the tissue’s functionality [30]. Characterization of this substitute revealed histological and functional features closely resembling native tissues [30]. The mechanical properties of the substitute were adequate for surgical handling, and its barrier function tests yielded results similar to those observed in native tissues [30].

The present study aimed to assess the in vivo performance of this human-derived tissue-engineered urethral substitute using the most-used animal model for urethral reconstruction, the adult New Zealand male rabbit [1,31]. Indeed, rabbits account for over 73% of preclinical models used to investigate urethral reconstruction [31]. Following clinical guidelines, a two-stage urethral reconstruction was performed using the substitutes engineered using cultured human urothelial and fibroblastic cells. The engraftment of these urethral substitutes in immunosuppressed adult rabbits was evaluated in addition to the technical feasibility of correcting an induced penile anomaly, i.e., an artificial penoscrotal hypospadias. The group receiving the engineered substitute was compared to a positive control group, where the urethra was longitudinally opened down to the penoscrotal junction but not removed to perform urethroplasty in the second stage.

## 2. Results

### Macroscopic and Histological Characterization of the In Vitro Engineered Substitutes and Native Urethral Tissues

Urethral substitutes reconstructed by tissue engineering were produced in vitro using the self-assembly technique as previously described [30]. Their macroscopic appearance (Figure 1A) showed homogenous tissues with a regular epithelium. The histological characterization is shown in Figure 1B for tissue-engineered substitutes and in Figure 1C for the native rabbit tissue. The epithelium of the substitutes presents the basal layer laid on the basal lamina, the elongated intermediate cells, and finally a layer of flat superficial cells (umbrella cells) [32]. Six rabbits received such substitutes (group 1) whereas six rabbits had their urethra incised to create a surgically induced hypospadias (group 2).

Two-stage urethroplasties were performed using the tissue-engineered substitutes (group 1). After the urethral section to be resected was identified, the substitute was implanted with a urethral stent left in place for seven days (first stage). Twelve weeks after surgery, the graft was tubularized (second stage). Eight weeks later, a urethroscopy was conducted, and healing continued for up to 28 weeks. Finally, following a terminal urethroscopy, the animals were subjected to a urethrographic study, and tissues were harvested for post-mortem analyses (Figure 2). In group 2, the native urethra was ventrally incised but not removed, with the meatus repositioned at the penoscrotal junction. Twelve weeks after surgery, the urethral plate was tubularized, and the group 2 rabbits followed the same protocol as the rabbits in group 1. As seen in the photographs at 28 weeks post-op (Figure 2), surgical defects were successfully corrected in most of the rabbits in group 1 and all rabbits in group 2.

Indeed, two out of six rabbits with engineered substitutes in group 1 showed no complications (Table 1). Unfortunately, two rabbits from group 1 were euthanized for issues unrelated to the urethral surgeries performed (specifically, ileus and dental malocclusion causing feeding difficulties). However, a rabbit in group 1 exhibited recurrent meatal stenosis at the penoscrotal junction and was euthanized prior to tubularization, and another rabbit presented recurrent urethral stenosis at the penoscrotal junction (the junction between the tissue-engineered substitute and the native urethra) and was euthanized before undergoing urethroscopy. No complication was noted in the surgical control group (group 2) until the end of the experiment (Table 1).

To monitor the healing of the urethras, urethroscopies were conducted at both 8 and 16 weeks following urethral tubularization (Figure 3A). For 2 rabbits among the 4 (excluding the 2 rabbits euthanized for reasons unrelated to the study), excellent graft integration was observed, with urothelium resembling that of the surgical control group (Figure 3B,C).

The urethrography demonstrated that 2 rabbits among 4 (without unrelated unwanted events) have an efficient reconstruction with a flow rate of liquid similar to the control group (Figure 3D,E). These results confirmed the macroscopic observations shown in Figure 2.

The quality of the tissues 28 weeks post-implantation was assessed. Post-implantation, samples from the group 1 (tissue-engineered urethral substitutes) and group 2 (surgical control) repaired sections were evaluated following Masson’s trichrome or Hematoxylin-Eosin staining (Figure 4). A urothelium was visible in the samples from the rabbits of group 1 (Figure 4A,B—Masson’s trichrome staining and Figure 4E,F—Hematoxylin and Eosin) and in the samples from the rabbits of group 2 (Figure 4C,D—Masson’s trichrome staining and Figure 4G,H—Hematoxylin and Eosin). Adequately differentiated urothelium was clearly visible in group 1 (Figure 4B,F vs. Figure 4D,H) with well-defined basal, intermediate, and superficial layers. The aspect of the stroma seems roughly similar between samples from rabbits of group 1 and group 2, especially extracellular matrix density (collagen stained in blue in Masson’s trichrome staining) and vascularization (pinkish structures in the samples stained by Masson’s trichrome staining protocol). No clear differences were noted between the aspect of the tissue-engineered matrix and the aspect of the native matrix 28 weeks after implantation. It can be noted that the general morphology of the tissue-engineered substitutes after implantation and tubularization remained round (Figure 4A,E), whereas a more flat morphology was observed in the urethra of the animals of the surgical controls (Figure 4C,G).

Blood vessels of varying sizes containing red blood cells were visible near the urethras of both group 1 (urethra repaired using the tissue-engineered substitutes) and group 2 (surgical control urethras) animals (Figure 5).

To ensure the tissue that was explanted was indeed the substitute engineered using human cells, immunolabeling was performed. The presence of human epithelium was assessed through immunolabeling using specific markers. The pancytokeratin AE1/AE3 antigen was utilized to stain the epithelium, while human leukocyte antigen (HLA) was employed to stain the human cells. AE1/AE3 staining was observed in both the reconstructed and native urethra 28 weeks after implantation (Figure 6A,B). While a strong HLA-related signal was detected in the engineered urethra (Figure 6C,E), as expected, no human HLA signal was detected in the native urethra (Figure 6D,F). HLA and epithelial antigens were identified in rabbits grafted with human-derived substitutes from group 1, 28 weeks post-implantation.

## 3. Discussion

Previous research has outlined the in vitro production protocol for engineered human urethral substitutes, which utilizes, in the context of the self-assembly approach, a combination of dermal fibroblasts (DF) and vesical fibroblasts (VF) to produce the stroma and urothelial cells for the epithelium [30]. Characterization of these substitutes revealed that their mechanical properties were adequate for surgical manipulation and provided a barrier function comparable to that of native tissue in vitro [30].

In the current study, the potential of these engineered human urethral substitutes was evaluated in vivo, using adult New Zealand white rabbits to reconstruct a segment of 2 cm of their urethra. A two-stage protocol was used. The first stage is the urethral removal and replacement by the tissue-engineered substitutes (group 1, 6 rabbits) or the opening of the urethra to create a surgically induced penoscrotal hypospadias (group 2, 6 rabbits). Approximately 12 weeks later, the urethra was tubularized, and anastomosis of the engineered substitutes to the native proximal urethra was performed. During the experiment, the provisional meatus allowed urine exposition of the tissue-engineered substitute to assess the engineered substitutes’ viability against cytotoxic urine as a replacement tissue [33].

In this study, group 1 exhibited 33% (2 rabbits among 6) of urethroplasty-unrelated unwanted events (ileus and malocclusion), 33% (2 rabbits among 6) had urethral complications (stenoses), and 33% had complete success or 50% if rabbits euthanized for study-unrelated reasons were ignored, and group 2 exhibited a complete success with 6 rabbits among 6 that completed the experiments. The grafted engineered substitutes showed a good integration documented by macroscopic aspects (Figure 2), urethroscopy (Figure 3B), and urethrography (Figure 3D) imaging and standard histology (Figure 4A,B,E,F). No signs of fibrosis were detected (Figure 4A,B). Additionally, there was no infiltration of lymphocytes observed in the histological cross-sections stained with Hematoxylin and Eosin (Figure 4E,F), indicating the absence of inflammatory response near the graft [34]. This is in line with an efficient tacrolimus immunosuppression allowing grafting of substitutes engineered using human cells.

Despite these promising results, some observations raised concerns. With the loss of 2 rabbits among 12, the attrition rate was 16.7%. It is not clear if this rate was attributable to the tacrolimus treatment. Nevertheless, force-feeding rabbits with tacrolimus remains a long, expensive, and unpleasant process for the animal. Recently, a genetically modified strain of immunosuppressed rabbits has been described [35] and could be used for subsequent experiments in order to decrease the attrition rate due to immunosuppression-related complications to an acceptable level.

The reconstructed urethra exhibited a circular shape (Figure 4A,E), in contrast to the native urethra, which is typically elongated or star-shaped (Figure 4C,G). The star-like shape is known to accommodate increased urethral pressure during urination, with studies indicating that star-shaped collagen scaffolds can withstand pressures up to 132 ± 22 mmHg, while circular scaffolds resist only 52 ± 21 mmHg [36]. Although the circular shape did not appear to hinder the functionality of the grafted tubes in this study, producing a star-shaped substitute may be beneficial for improving burst pressure. However, a star shape was observed when the reconstructed urethras were anastomosed to the native urethra (e.g., Figure 4B), likely due to active remodeling by the surrounding native tissues, suggesting that a star-shaped substitute may not be essential on the day of implantation.

Another issue of this study is related to the choice of the rabbit model. Although rabbits are a commonly used model in urethroplasty research, their tissue contractility differs from that of humans. Rodents tend to heal wounds through contraction, potentially up to 80%, while human wounds typically heal through re-epithelialization [37]. For instance, Pinnagoda et al. used an acellular collagen scaffold to address a 2 cm urethral deficit in a rabbit model, reporting 20% incidence of stenosis, 20% of fistulae, and 60% with no complications [38]. In another study, Arenas da Silva et al. employed a tubular collagen I scaffold, with seeded scaffolds showing 25% stenosis and 25% fistula, while unseeded scaffolds exhibited 100% stenosis [39].

In our current study, urethral grafts from group 1 exhibited significant graft contraction shortly after implantation, emphasizing the need to consider anatomical and behavioral differences between rabbits and humans, which likely complicate graft healing. Notably, the rabbit’s penis remains internalized at rest, which can create friction against the grafted tissue, complicating graft maintenance and healing. This retraction could result in graft folding, hindering adherence to the graft bed and leading to vascularization deficits, ischemia, and ultimately, contracture. Unlike human surgeries, applying compressive dressings on the rabbit’s penis poses a challenge. Additionally, when internalized, the graft is subjected to a warm, humid environment, which can impede adequate and rapid healing. Post-surgery, the use of an Elizabethan collar for five days helped prevent rabbits from scratching the graft. However, this caused physical discomfort that hindered eating and grooming. Once the collar was removed, rabbits tended to scratch the graft site, sometimes attempting to tear it off.

In these experiments, it is difficult to determine whether the contraction primarily stems from the graft material itself or the host rabbit tissue, as similar results were noted in the clinical control group. To address this issue, alternative animal models with reduced tissue contractility during healing, such as pigs [40], could be employed, or surgical techniques could be modified. For instance, tubularizing two flat tissues around a tube or implanting a tubular construct subcutaneously adjacent to the urethra as a first stage could be a viable solution instead of conducting a regular two-stage surgery. Other alternatives might include producing larger tissue surfaces and allowing them to contract overnight before grafting or performing a pre-vascularization step [41].

## 4. Materials and Methods

### 4.1. Ethics Statement

This work was conducted according to the Declaration of Helsinki. It was approved by the institution’s committee for the protection of human participants (Comité d’éthique de la recherche du CHU de Québec-Université Laval, protocol number 2012-1341). All donors provided informed written consent before biopsies. The experimental animal protocol 2021-748-CHU-21-748 was approved by the Ethics Committee of Laval University (Université Laval) affiliated with the university hospital (CHU de Québec).

### 4.2. Cell Culture

Vesical fibroblasts (VF) and urothelial cells (UC) were isolated from human bladder biopsies, whereas dermal fibroblasts (DF) were isolated from human skin biopsies, as previously described [42,43,44]. Biopsies were obtained during reconstructive surgeries for benign conditions. All biopsies came from male donors (distinct donors for each cell type, two donors for bladder biopsy, and one donor for dermal biopsy). Cultured cells were stored frozen in liquid nitrogen until thawing before use. They were used between passages 2 and 4. Epithelial cells were cultured in UC medium containing a 3:1 mix of Dulbecco–Vogt modification of Eagle’s (DMEM, Invitrogen, Burlington, ON, Canada) and Ham’s F12 (Flow Lab., Mississauga, ON, Canada) supplemented with 5% fetal bovine serum (FBS-H) (GE Healthcare, Chicago, IL, USA), 24.3 μg/mL adenine (Sigma-Aldrich), 5 μg/mL crystallized bovine insulin (Sigma Aldrich, St. Louis, MO, USA), 1.1 μM hydrocortisone (Teva Canada Ltd., Scarborough, ON, Canada), 0.212 μg/mL isoproterenol hydrochloride (Sandoz Canada, Boucherville, QC, Canada), 10 ng/mL epidermal growth factor (Austral Biologicals, San Ramon, CA, USA), and antibiotics: 100 U/mL penicillin and 25 mg/mL gentamicin (Sigma-Aldrich). Fibroblasts were cultured in a medium made of DMEM supplemented with 10% fetal bovine serum (FBS) (Invitrogen) and antibiotics.

### 4.3. Production of Engineered Tridimensional Flat Urethral Substitutes

The tridimensional urethral substitutes were produced using a modified method derived from the self-assembly technique called the hybrid technique [45,46]. Briefly, 80% of vesical fibroblasts and 20% of dermal fibroblasts were seeded at a density of 5.2 × 10^4^ cells/cm^2^ in 6-well plates (Falcon, ThermoFisher, Waltham, MA, USA) and cultivated in Dulbecco-Vogt modification of Eagle’s medium (DMEM, Invitrogen, Burlington, ON, Canada) containing 10% fetal bovine serum (Hyclone, Logan, UT, USA), 100 U/mL penicillin, and 25 mg/mL gentamicin (Fb medium) supplemented with 50 μg/mL ascorbate (Sigma-Aldrich). A second stromal cell seeding was performed at the same density on day 14 on the newly formed stromal sheets. The culture was maintained until enough newly synthesized ECM had accumulated to form a manipulable cell sheet, typically achieved within 28 days. At this point, three stromal sheets were superimposed, and UCs were seeded on top of the tridimensional tissues at a density of 5.2 × 10^4^ cells/cm^2^. Tissues were cultured with UC medium supplemented with 50 μg/mL ascorbate. After seven days under submerged conditions, the flat urethral tissues were elevated at the air/liquid interface for 21 days to induce the maturation of the urothelium. The tissues were either grafted to animals or subjected to histological analyses.

### 4.4. Animal Experiments

Animal experiments were designed according to the ARRIVE guidelines.

Study design: In the experiment, the rabbits were divided into two groups: Group 1 included the rabbits receiving the tissue-engineered substitutes (group 1 = reconstructed urethra group), whereas the rabbits in group 2 received autologous rabbit urethral tissues to repair the surgically induced hypospadias (group 2 = surgical control group).

Rabbits were kept in individual cages that were ventilated in a temperature-controlled room (21 ± 1 °C) with a cycle of 12 h of light and 12 h of darkness. Rabbits had access to water and hay ad libitum. They also received a controlled number of pellets. Fruits and vegetables were provided as additional nutritional supplements and enrichment for the rabbits. As part of enrichment, rabbits had access to playtime in an enclosure twice a week.

Sample size: Sample size of each group: 6.

How was determined the sample size: Sample size calculation: Five animals are required at the end of the 9-month observation period for statistical significance, and we expect up to 20% attrition over 9 months; therefore, we need six rabbits per study group at the beginning. This sample size will achieve 80% power for a non-inferiority test of two independent proportions, assuming a 35% non-inferiority margin (confidence interval: 0–35%) of failure (urethral stenosis) for the experimental groups and 20% for the gold standard control group, with alpha set at 0.05.

Randomization: The animals were distributed randomly into the 2 groups.

Outcome measures: The outcome of the experimental procedure was monitored by cystoscopy (2 times) and urethrography (at the end-point). Also, macroscopic aspects and histological organization of the urethra were evaluated. The presence of a human-derived epithelium was confirmed by immunofluorescence against the HLA.

Experimental animals: Twelve 12-month-old male adult New Zealand white rabbits weighing between 2.5 and 3 kg were included in this study (Charles River Laboratories, Senneville, QC, Canada).

#### Experimental Procedures

Immunosuppression

Animal immunosuppression was performed through Tacrolimus considering the human origin of the engineered substitutes [47]. Tacrolimus (FK506, Sigma-Aldrich, Mississauga, ON, Canada) solutions were prepared by mixing 5 mg capsule powder with Simple syrups (Atla Laboratories, Montréal, QC, Canada) and Oral-Plus suspending vehicle (Perrigo, Minneapolis, MN, USA), for a final concentration of 1 mg/mL. A dose of 1.0 mg/kg was applied with no significant treatment-related side effects observed. The treatment of rabbits with 1 mg/kg of tacrolimus was initiated one week prior to surgeries and administered until the end of the protocol. To determine the efficacy of the immunosuppression, blood samples were collected before oral administration. All blood levels of tacrolimus remain in the efficient range during the experiment (i.e., 5 to 10 ng/mL, see Appendix A).

Surgeries

Rabbits underwent surgery under general anesthesia using 5% isoflurane (Abbott, Markham, ON, Canada). To ensure pain management, buprenorphine (0.05–0.1 mg/kg, Champion Alstoe, Whitby, ON, Canada) was administered preoperatively, followed by daily doses of carprofen (5–10 mg/kg, Pfizer, Kirkland, QC, Canada) for 48 h post-surgery. An Elizabethan collar was fitted for five days following surgery to prevent self-injury. All surgical procedures were performed under sterile conditions, and mechanical ventilation was used for the duration of the surgery.

Group 1 (6 rabbits) was operated on as follows: a 1.5 cm incision was made in the penile skin on the ventral side, followed by the immediate removal of a 2 cm segment of the native urethra. The human-engineered urethral substitute was then implanted to cover the defect and sutured to the skin edges using absorbable 7-0 polydioxanone sutures, mimicking what would be performed during a two-stage urethroplasty procedure in clinical practice. The urethral meatus was repositioned at the penoscrotal junction.

After a ten-week healing period, both groups were evaluated to determine whether tubularization of the graft could be performed (second stage of the urethroplasty). Two weeks after the positive evaluation (a 12-week recovery), three edges of the graft (the two lateral sides and the proximal edge) were incised to create a vascularized flap. The right side of the graft was then sutured to the left side, extending to the proximal end, while leaving the distal end open to form a tube with an orifice resembling the meatus. The newly formed urethra was proximally anastomosed to the native urethra at the penoscrotal junction, with the distal end forming the new meatus. The skin was closed over the newly formed urethral tube using fast-absorbing 5-0 Vicryl^TM^ sutures. All procedures were carried out by the same urologist with expertise in urethral reconstruction.

Group 2 (6 rabbits), the surgical control group, underwent surgery under general anesthesia and analgesia, as previously described. A 1.5 cm incision was first made along the ventral side of the penile skin. Then, the distal 2 cm of the native urethra was ventrally incised but not removed, with the meatus repositioned at the penoscrotal junction. The edges were sutured to the skin edges using absorbable sutures.

After a ten-week healing period, an assessment was conducted to determine if urethral tubularization could be performed two weeks later (following 12 weeks of recovery). For the tubularization procedure, the edges of the incised native urethra were freed and sutured back together using 7-0 absorbable sutures. The skin was then closed over the urethra using rapid-absorbing 5-0 Vicryl sutures. All procedures were performed by the same urologist.

### 4.5. Urethroscopy and Urethrographic Studies

Retrograde urethroscopy and urethrography were performed 8 and 16 weeks after tubularization. To do so, a 3 mm diameter (9 Fr) urethroscope was connected to a camera to inspect the urethra from the meatus to the proximal urethra, emphasizing the repaired site where the entire mucosal circumference was inspected. Images were acquired using an iPhone 8 (Apple, Cupertino, CA, USA). Retrograde urethrography assessments were performed in the “Institut Universitaire de Cardiologie et de Pneumologie de Québec animal facility” (IUCPQ) within hours after euthanasia to evaluate the presence of potential strictures or fistulas. An 8F catheter was placed in the urethral orifice connected to a syringe containing the contrast agent iodixanol 320 mg/mL Visipaque (GE Healthcare, Chicago, IL, USA). X-ray images were obtained using Azurion 7 C12 (Philips, Cambridge, MA, USA) after injection of the contrast agent into the urethra in a supine position. Fluoroscopic images were treated using MicroDicom DICOM viewer 2022.1 software (Sofia, Bulgaria).

### 4.6. Histological Analyses

The flat-engineered human urethral substitutes were produced in larger quantities than required for grafting. These substitutes were then fixed using 3.7% formaldehyde, embedded in paraffin, sectioned at 6 μm, and stained with Masson’s trichrome (MT) or Hematoxylin and Eosin (H&E). Images were captured with a Zeiss Axio Imager M2 microscope equipped with an AxioCam ICc1 camera (Oberkochen, Germany).

### 4.7. Immunolabeling

OCT-embedded cryosections (10 μm thick) of urethral tissues were processed for immunofluorescence (IF). AE1/AE (Millipore-Sigma, MAB3412, St. Louis, MO, USA) and HLA (Abcam, ab70328, Cambridge, UK) primary antibodies were respectively used at the dilution 1/400 and 1/200, and cell nuclei were stained with Hoechst solution (Sigma). At least ten tissue sections of each substitute (three per experimental group) were analyzed under a Nikon Eclipse E600 epifluorescence microscope (Nikon, Mississauga, ON, Canada).

## 5. Conclusions

The tissue-engineered urethral substitutes produced in vitro using human cells facilitated the reconstruction of an artificial urethra. The construct provided functional characteristics prior to implantation (good mechanical resistance and a mature urothelial coverage). Despite our efforts, the strategies employed resulted in the failure of the non-inferiority test based on literature-established parameters. Specifically, the success rate was 33%, increasing to 50% when urethroplasty-unrelated events were excluded. This remains below the expected lower limit of success set at 65%. We hypothesize that this failure is partially attributable to the side effects of tacrolimus for the urethroplasty-unrelated events and the unique penile anatomy of rabbits, where penile retraction may cause graft damage through friction, folding, and delayed graft take. Modifications to the experimental protocol will be necessary to improve the success rate. Future research could consider tubularizing the construct around a supportive material (such as agarose or silicone), with the urothelium facing inward, before implantation subcutaneously in rabbits, adjacent to the urethra, acting as a first-stage procedure. By implanting the graft under the skin first, this approach would help prevent damage to the graft by the rabbits. In the second stage, the native urethra could be excised and replaced with the adjacent reconstructed tubular urethra.

Larger urethral tissues can be produced using the same techniques as described in Kawecki et al. [47]. It is also possible to produce tubular structures in vitro if necessary, as we have carried out previously [41,48]. From a long-term perspective, i.e., in clinical studies, tissues should be produced in a Good Manufacturing Practice (GMP) environment. This involves the use of clean rooms and modification of practices, including potentially changing protocols (as performed for skin [49] or cornea [50] using the same technique, i.e., self-assembly) and reagents (e.g., using a serum-free medium such as that developed by Caneparo et al. [51]). The cost of the product will increased but would remain within an acceptable range.

The use of tissue-engineered urethral substitutes represents a promising avenue for urethroplasty. Nevertheless, adjustments to the pre-clinical protocol are needed to address the challenges encountered with this rabbit model.

## Figures and Tables

**Figure 1 ijms-26-01825-f001:**
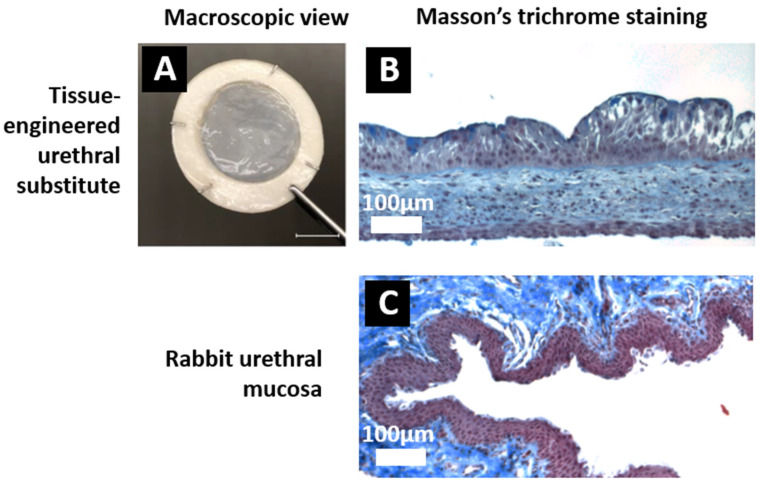
Evaluation of the tissue-engineered substitutes before implantation. Urethral substitutes reconstructed by tissue engineering have been produced. (**A**) Macroscopic view of the substitutes (Scale bar: 1 cm). (**B**) Tissue-engineered substitute samples were evaluated histologically after Masson’s trichrome staining (Scale bar: 100 μm). (**C**) Native urethral mucosa of a rabbit stained using Masson’s trichrome (Scale bar: 100 μm).

**Figure 2 ijms-26-01825-f002:**
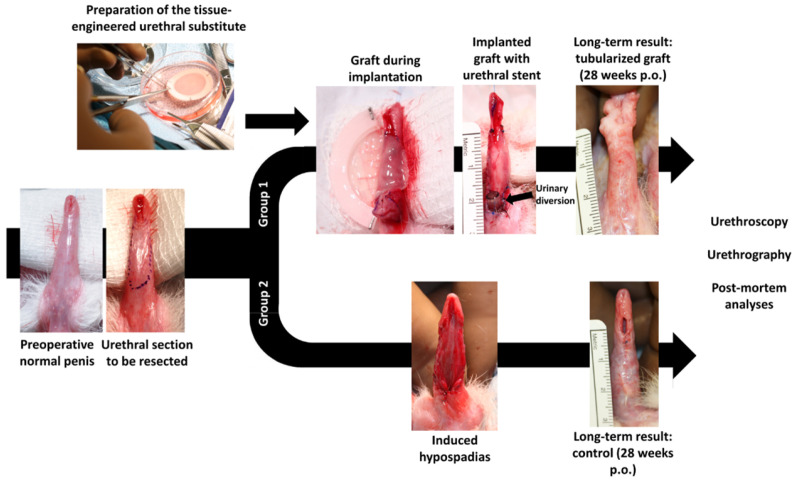
Experimental design. Photographs were taken to illustrate the successive steps of the study. The human-derived tissue-engineered substitutes were produced in vitro and prepared for the surgeries. The section to be resected was identified. Rabbits were then divided into two groups: group 1 included the rabbits receiving the tissue-engineered substitutes, whereas the rabbits in group 2 received autologous rabbit urethral tissues to repair the surgically induced hypospadias. Representative photographs illustrate the long-term results of the reconstruction using substitutes or native rabbit tissues (28 weeks).

**Figure 3 ijms-26-01825-f003:**
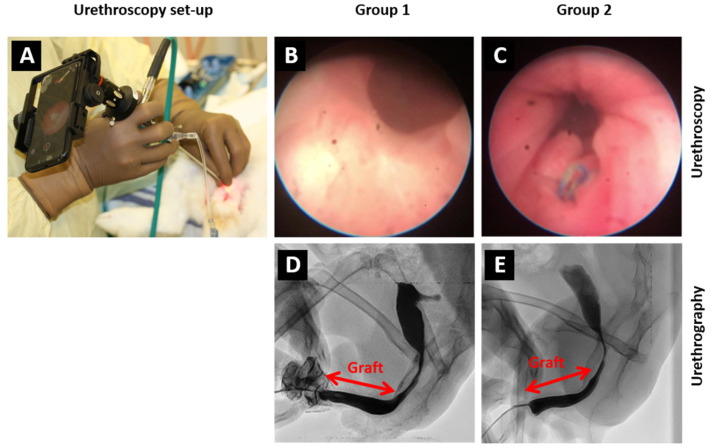
Urethroscopy and urethrographic studies. (**A**) Urethroscopy set-up, (**B**) Urethroscopy of the reconstructed rabbit’s urethra grafted with engineered urethral substitutes (group 1) or (**C**) for the surgical control group (group 2). (**D**) Urethrography of the reconstructed rabbit’s urethra grafted with engineered human urethral substitutes (group 1) or (**E**) for the surgical control group (group 2). Representative images are presented.

**Figure 4 ijms-26-01825-f004:**
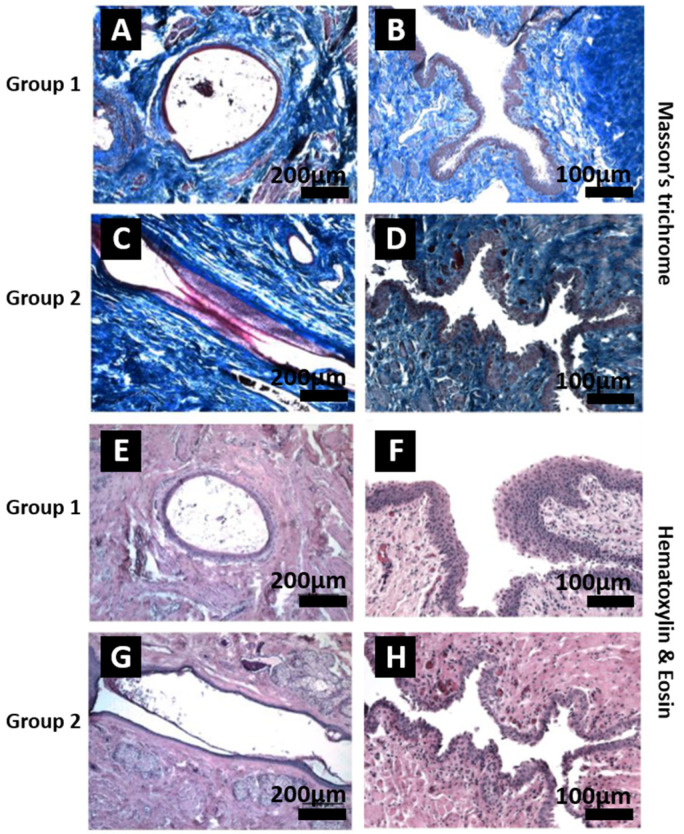
Post-implantation histological evaluation of the reconstructed urethra. Urethras were evaluated following Masson’s trichrome staining (**A**–**D**) or Hematoxylin and Eosin staining (**E**–**H**) and carried out in triplicate. Urethras repaired using the tissue-engineered substitutes (group 1) are presented in (**A**,**B**,**E**,**F**), whereas surgical control urethras (group 2) are presented in (**C**,**D**,**G**,**H**). Scale bars are 200 µm for (**A**,**C**,**E**,**G**) and 100 μm for (**B**,**D**,**F**,**H**).

**Figure 5 ijms-26-01825-f005:**
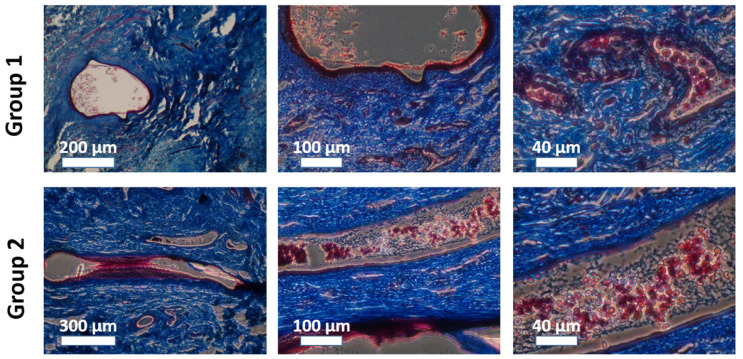
Post-implantation histological evaluation of the vascularization near reconstructed urethra. Vascularization near urethras was evaluated following Masson’s trichrome staining for group 1 (urethra repaired using the tissue-engineered substitute) and group 2 (surgical control urethras). Sequential increasing magnifications (lower on the left and higher on the right) are presented to allow better visualization of red blood cells in small blood vessels near urethras. Scale bars are as indicated.

**Figure 6 ijms-26-01825-f006:**
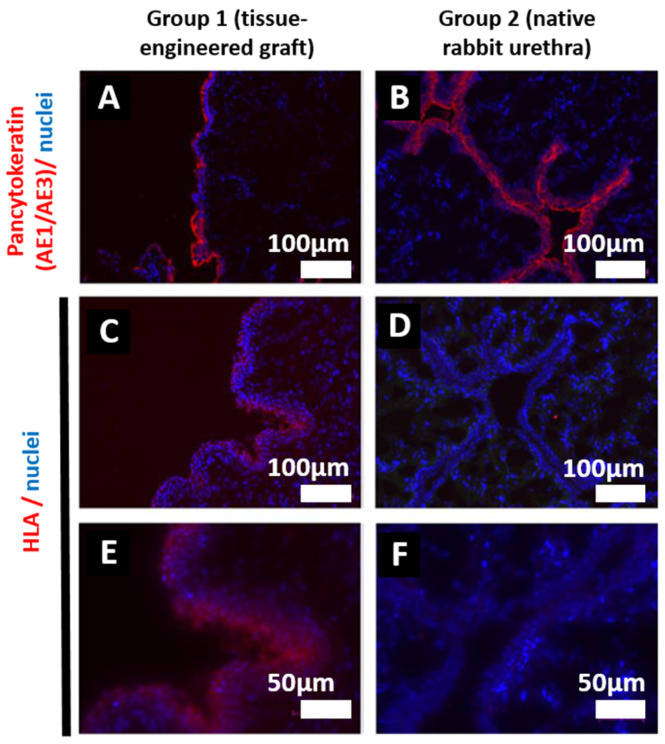
Immunolabelling characterization of the reconstructed urethra 28 weeks after implantation. Reconstructed urethras using urethral substitutes (group 1, (**A**,**C**,**E**)) and urethras reconstructed with native tissue (group 2, (**B**,**D**,**F**)) were subjected to immunolabeling to confirm the human origin of the analyzed grafted tissues. Representative images from tissues (in triplicates) were taken using a fluorescence microscope: Labeling of AE1/AE3 antigens, identifying the epithelial cells (**A**,**B**). Labeling of HLA antigens at the surface of human tissues (**C**–**F**). (**E**,**F**) show higher magnification of the HLA immunostaining. (**A**–**D**) Scale bar: 100 μm. (**E**,**F**) Scale bar: 50 μm.

**Table 1 ijms-26-01825-t001:** Follow-up of animals.

Complications	Group 1 (Tissue-Engineered Substitutes; 6 Rabbits)	Group 2 (Surgical Control; 6 Rabbits)
Adverse side effects related to urologic tissues	2 (stenoses)	0
Adverse side effects unrelated to urologic tissues	2 (1 ileus, 1 malocclusion)	0
No complications	2	6

## Data Availability

All data are available upon reasonable request from the corresponding author.

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
