# Peer review of "Correction of Significant Urethral Anomalies Using a Tissue-Engineered Human Urethral Substitute: Proof of Concept"

_ijms, 2025, doi:10.3390/ijms26051825_

Round 1

Reviewer 1 Report

Comments and Suggestions for Authors

I have read with interest an article entitled: “Correction of significant urethral anomalies using a tissue-engineered human urethral substitute: proof of concept.” The topic of the paper is important because in literature, still a small number of articles related to this topic, especially performed on animal models, were published. In order to improve manuscript quality some changes have to be performed:

1.      Figure 5. In my opinion, molecular characterization was used inappropriately. It is an immunofluorescence characteristic of urothelium regeneration.

2.      In chapter 4.4 Author wrote that eighteen 12-month-old rabbits were used, while in the study description, two groups were used (6 animals in each). What happens with the rest of the 6 rabbits?

3.      On page 8, line 206 Authors wrote about lymphocyte infiltration. In order to analyze lymphocytes, appropriate immunohistostaining should be performed. Because tacrolimus was used throughout all experiments, the analysis of the inflammatory response is controversial.

4.      Discussion about the possibility of creating larger substitutes for clinical purposes and the costs associated with the need to adapt to GMP standards should be performed.

5.      The main problem is the lack of statistically significant importance of obtained results. In the study group, only in two cases (33%) successful reconstruction was noticed; in other tested animals, side effects (related and non-related to the procedure) were noticed. In the control group, all procedures were successful. More animals and another model that will eliminate the need for immunosuppression should be used.

Reviewer 2 Report

Comments and Suggestions for Authors

A detailed report can be found in the attached document.

Round 2

Reviewer 1 Report

Comments and Suggestions for Authors

The authors answered almost to all my questions. The problem of the lack of statistically significant importance of obtained results, in my opinion, was not properly addressed. Authors wrote: This sample size will achieve 80% power for non-inferiority test of two independent proportions, assuming a 35% non-inferiority margin (confidence interval: 0-35%) of failure (urethral stenosis) for the experimental groups, and 20% for the gold standard control group, alpha set at 0.05; however in the manuscript, no statistical test was used to compare results between test and control group. Because only in two animals in the experimental group successfully reconstruction was noticed, a margin of failure was higher than expected (77%). That is why explanations regarding the effectiveness of the proposed method, preferably supported by calculations, should appear.

Round 3

Reviewer 1 Report

Comments and Suggestions for Authors

I agree with the Authors that negative results can sometimes provide useful knowledge. The authors should add in the Abstract information that the results obtained for the experimental group were less effective compared to the control group and maybe that the animal model used was not suitable and should be changed. In the current abstract form, the reader may get a false impression of the actual results.
